# A CEEMDAN-Assisted Deep Learning Model for the RUL Estimation of Solenoid Pumps

Ugochukwu Ejike Akpudo 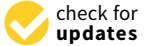 and Jang-Wook Hur *

Department of Mechanical Engineering, Department of Aeronautics, Mechanical and Electronic Convergence Engineering, Kumoh National Institute of Technology, 61 Daehak-ro (Yangho-dong), Gumi 39177, Gyeongbuk, Korea; akpudougo@gmail.com
* Correspondence: hhjw88@kumoh.ac.kr

**Abstract:** This paper develops a data-driven remaining useful life prediction model for solenoid pumps. The model extracts high-level features using stacked autoencoders from decomposed pressure signals (using complementary ensemble empirical mode decomposition with adaptive noise (CEEMDAN) algorithm). These high-level features are then received by a recurrent neural network-gated recurrent units (GRUs) for the RUL estimation. The case study presented demonstrates the robustness of the proposed RUL estimation model with extensive empirical validations. Results support the validity of using the CEEMDAN for non-stationary signal decomposition and the accuracy, ease-of-use, and superiority of the proposed DL-based model for solenoid pump failure prognostics.

**Keywords:** empirical mode decomposition; CEEMDAN; deep feature learning; stacked autoencoders; gated recurrent units; remaining useful life estimation

## 1. Introduction

Modern engineering practice is showing a strong dependence on artificial intelligence (AI) for virtually all state-of-the-art solutions. Among these solutions is the use of bio-inspired mathematical models with deep architecture machine learning (ML) and deep learning (DL) methods for diagnostic and prognostic purposes [1,2]. Beyond the limitations of traditional ML techniques, DL techniques come with diverse advantages including big data compatibility, automated feature engineering, and ease-of-use; however, traditional ML techniques like support vector machines, decision tress, logistic regression, etc. retain their superiority on small data and low-end hardware. Basically, the success of state-of-the-art data-driven predictive maintenance frameworks rely greatly on the effectiveness of the DL-based diagnostic and/or prognostic algorithm at its core. This presents ample opportunities for developing (and improving) high-performing generalized models for accurate failure diagnostics and prognostics, even for unseen equipment working conditions [3]. Particularly for the remaining useful life estimation of equipment/components, several articles have recorded various successes in the use of DL methods regardless of the types and architecture of the artificial neural network (ANN) employed [4].

Generally, the success of DL methods for enhanced high-level transient and spectral feature extraction from non-stationary signals and automated feature learning cannot be overemphasized, particularly in this era of increasing demand for optimum reliability from predictive maintenance algorithms on one hand and the availability of computational resources on another hand. Their superior capabilities against the numerous traditional statistical approaches for hand-crafted feature extraction are quite commendable since they are capable of extracting both discriminative and prognosible features at multiple levels [5,6]. For instance, with the aid of convolving filters and deep multi-layers, convolutional neural networks (CNNs) are capable of deep feature learning in *supervised* cases—image recognition [7], fault diagnostics [8], anomaly detection, etc. [9]. This makes them reliable stand-alone fault diagnostic tools. On the other hand, sparse autoencoders are

robust for deep feature learning in *unsupervised* cases as presented in this study. More so, recurrent neural networks (RNNs) are also popular for learning transient dependencies, and with the recent advancement in their traditional designs, state-of-the art RNN variants—long short term memory (LSTM) and GRU neural networks are capable of long–range memorization for accurate posterior estimations, with the latter more efficient than the former [10]. Stacked autoencoders (SAE) provide automated feature representation (learning) capabilities from inputs; however, their efficiencies for identifying transient inputs are limited [11]. This deficiency in capturing transient dependencies can be compensated by employing a GRU which although similar to LSTMs, are less computationally expensive.

Generally, data-driven predictive maintenance relies significantly on the availability and prognosibility of sensor data. This usually demands the use of multiple sensors and/or multiple sensor-extracted features (in the case of a single-sensor situation) [4]. Because most sensor measurements are non-stationary (with some background noise), signal de-noising is almost a necessary pre-processing step to ensure that noise and/or insignificant spectral components of the signals are excluded from the feature extraction and dynamic modelling process. Interestingly, several sparsity-based and Bayessian filter-based de-noising techniques have been developed over the years with remarkable efficiencies across diverse applications. Among these is the empirical mode decomposition (EMD) which decomposes signals into a series of complete orthogonal intrinsic mode functions (IMFs) based on the local characteristic transient information in the signal [12]. Through the Hilbert Huang transform (HHT), these finite IMFs produce instantaneous frequencies with practical significance.

As a result, integrating a de-noising module for a more reliable feature learning does not only ensure better prognostic results, but it also provides a verifiable standpoint from which empirical validations can be drawn. To better understand this study, this paper is structure thus: Section 2 discusses recent related works on DL-based prognostics and the motivation for our proposed study while Section 3 introduces the proposed CEEMDAN-assisted SAE-GRU prognostics model (and its dependencies). Section 4 presents the experimental case study and results while the paper is concluded in Section 5.

## 2. Related Works

Sudden equipment failure remains a major challenge on profitability in industries. In the quest for preventing this, the costly routine-based maintenance schemes seem outdated and has motivated the development and integration of predictive maintenance modules into production activities [2]. Ongoing research on reliability studies suggests that hydraulic components are some of the most complex systems to accurately monitor and has motivated diverse prognostic and health management (PHM) methodologies for these components—pumps, valves, actuators, etc. This complexity is a result of the thermodynamic, fluid dynamic, mechanical, and electrical processes (not to mention uncertainties) which simultaneously occur as the components are in operation [13]; nonetheless, vibration monitoring, acoustic emission monitoring, and pressure monitoring are among the most popular methods for pumps [4,14].

Solenoid pumps function by magnetization of the solenoids when electrical current passes through the coil which causes the electromagnetic core to move against a spring to slide a diaphragm into the discharge position. Consequently, the delivery (through a nozzle) and suction pressures/flow-rates better reflect its operational performance so a deviation from the ideal operating condition would be captured in the pressure measurements. Notwithstanding the durability of pressure monitoring for solenoid pumps, issues associated with sensor installation, calibration and signal conditioning result in the inevitable corruption of sensor measurements by background noise. This prompts the need for de-noising as a pre-processing step for accurate diagnostics/prognostics. Against the limitations of envelope analysis which relies on the assumption of signal stationarity and incognizance of transient information [15], and the discrete wavelet decomposition technique which is limited by the choice of base wavelet [16], the EMD technique reserves

its robustness for signal decomposition from which the IMFs provide reliable instantaneous frequencies with practical significance [12]; however, in the quest for solving the resulting mode-mixing problem caused by intermittent signals and minimizing the computational costs, several variations have been proposed including the ensemble EMD (EEMD) by Wu and Huang [17], the Complementary EEMD (CEEMD) developed by Yeh et al. [18], and the most efficient of the varants—CEEMDAN developed by Torres et al. [19]. The CEEMDAN further improves the decomposition efficiency of the EMD and the other variants by eliminating useless IMFs which are otherwise generated by the EMD and EEMD; thereby, improving the computation efficiency.

At incipient failure stages, pressure fluctuations start to develop rhythmic behaviours which typically contain several frequency components that vary across equipment components. This only further validates the need for *unsupervised* deep feature learning/extraction. For supervised feature extraction, CNNs are by far, the more reliable choice; however, the case presented in this study in an unsupervised case which is implied by the multiple *unlabelled* IMFs from the pressure signals. This presents an opportunity for the SAE to flourish since they are efficient for learning deep feature representations from multiple inputs [20]. The deep feature learning capabilities of SAEs have been recorded for many purposes including epileptic seizure detection [21], rotating machinery prognostics [6], anomaly detection [14], and a host of many other applications. In [21], the authors used SAEs to learn feature representations from multiple patient-specific scalp electroencephalograms (EEGs) which were fed into logistic classifiers for epileptic seizure detection in humans. Although the functionality of the SAEs for learning patient-specific seizure patters were well reported, a more robust diagnostic tool may have further minimized the false detection rate. Similarly, the authors of [6] proposed an explainable DL-based prognostic model in which a sparse autoencoder model was employed for high-level feature extraction from the Fourier transform of vibration signals which then constituted inputs for a feed-forward neural network for the RUL prediction of rotating machinery. On the other hand, an SAE-LSTM model was developed by the authors of [14] for detecting anomaly (fault) in mechanical equipment in cases whereby knowledge about anomaly is absent. The SAE feature learner accepts the outputs from a wavelet packet decomposition (WPD) process and provides the learned features to the LSTM predictor for anomaly detection. Results showed a 99% detection accuracy. Against the costs of these methods, the SAE-GRU comes with better computational efficiencies due to the GRU integration.

Undoubtedly, the use of deep hybrid prognostics methods have recently proven more reliable than traditional ML-based methods. Although the latter comes with strong theoretical and empirical validations against the former, they are usually associated with expensive assumptions, strenuous feature engineering procedures, and complex modelling processes. On the other hand, the availability of super computational resources in recent times has eased in the use of the more reliable DL methods for state-of-the-art predictive maintenance practice [22]. On the bright side, findings suggest that GRUs are much more efficient than the traditional time-series forecasting tools—RNNs and LSTMs with better advantages including solving vanishing gradient problems, computational advantages, better temporal modelling, and higher predictive accuracies [23,24].

## 3. Proposed CEEMDAN-Assisted SAE-GRU Model

The overall process for the RUL estimation is presented in Figure 1. The model receives pressure signals from the transmitters and is received by the fault detection module which is comprised by the CEEMDAN-based IMF extraction and SAE-based HI construction sub-modules, respectively. The output of the fault detection module (when a fault threshold is reached) is received by the GRU-based prognostics module for RUL estimation.

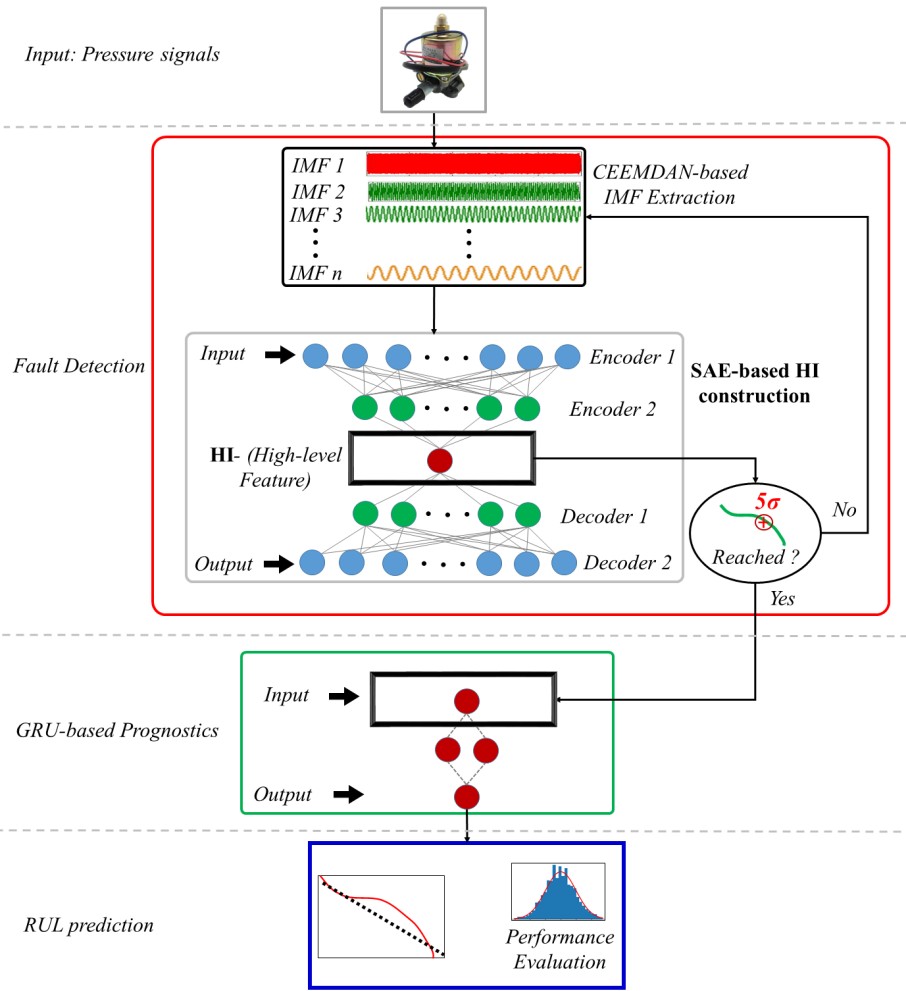

**Figure 1.** Proposed RUL prediction model.

The outputs from the CEEMDAN-based IMF extraction sub-module are fed into the SAE sub-module for deep feature extraction and health indicator (HI) construction. Following a successful training, a high-level single representation of the features is produced at the SAE's bottleneck which is then received by the GRU neural network for condition assessment/monitoring and RUL estimation. In detail, the following subsections provide the theoretical background of the constituent modules.

### 3.1. Fault Detection Module

Conventionally, in a reliable prognostics scheme, RUL estimation is by default, triggered once there is a deviation from normal/healthy state of health in a target system—when fault occurs. The comprehensive goal of our work is to model a high performing DL-based framework that can effectively and readily predict RULs of equipment/components with early automated warning capabilities. Invariably, its success relies on the effectiveness fault detection module consisted by the CEEMDAN-based IMF extraction, SAE-based HI construction, and the early warning/fault alarm sub-modules, respectively.

#### 3.1.1. CEEMDAN-Based Decomposition

Over two decades ago, the EMD was proposed by Huang et al. [25] for time-frequency signal processing. This algorithm outputs IMFs that are narrow-band components as representation of input signals using the Hilbert transform. Sadly, this traditional technique results to mode-mixing problems caused by intermittent signals and prompted the development of improved versions of the algorithm—EEMD and CEEMD—but because their computational process is dependent on the addition of white noise to the signal, the

overall cost of the decomposition is increased. Consequently, the CEEMD flourishes by suppressing the residue of the added white noise in EEMD [18].

Since EEMD and CEEMD both rely on EMD, they both suffer identical frequency resolution which ultimately results to inseparability problems between the relatively low frequency and relatively high frequency components even for a reasonable number of iterations. Fortunately, CEEMDAN offers superior solution by generating fewer IMFs on the premise of separating different signal components by using the two algorithms. The EEMD outputs the *true* modes as the average IMFs extracted from an ensemble of the original signal plus white noise with different strengths. Given a non-stationary signals $X(t)$, the CEEMDAN is computed from the EEMD by following the steps below:

1.  The mixed signals $X_i(t)$ are obtained as the sum:

$$X_i(t) = X(t) + \beta \omega^{(i)}(t) \tag{1}$$

    where $\beta$ is the variance of added white noise, and $\omega^{(i)}(t)$ $(i = 1, ...; i)$ is a zero-mean, unit variance white noises $N(0, 1)$

2.  Decompose each $X_i(t)$ using EMD to extract their first IMFs $d_k^i(t)$ $(k = 1, ..., N;)$ using (2).

$$\tilde{d}_1(t) = \frac{1}{N} \sum_{i=1}^{N} d_1^i(t) = \overline{d_1}(t) \tag{2}$$

3.  At stage $k = 1$, calculate the first residue using (3)

$$r_1(t) = f(t) - \tilde{d}_1(t) \tag{3}$$

4.  decompose the realizations

$$r_1(t) + \varepsilon_1 E_1\left(\omega^i(t)\right), i = 1, \ldots, N$$

    untill their first EMD mode, then compute the next mode using (4):

$$\tilde{d}_2(t) = \frac{1}{N} \sum_{i=1}^{N} E_1\left(r_1(t) + \varepsilon_1 E_1\left(\omega^i(t)\right)\right) \tag{4}$$

5.  The $k$th residue $(k = 2, ..., K)$ is generated by:

$$r_k(t) = r_{k-1}(t) - \tilde{d}_k(t) \tag{5}$$

6.  Decompose realizations

$$r_k(t) + \varepsilon_k E_k\left(\omega^i(t)\right)$$

    untill their first EMD mode is reached , and define the $(k + 1^{th})$ mode as:

$$\widehat{d_{(k+1)}}(t) = \frac{1}{N} \sum_{i=1}^{N} E_1\left(r_k(t) + \varepsilon_k E_k\left(\omega^i(t)\right)\right) \tag{6}$$

7.  Repeat steps 4–6 untill the obtained residue does not have at least two extrema. Consequently, the final residue becomes:

$$R(t) = f(t) - \sum_{k=1}^{K} \tilde{d}_k(t) \tag{7}$$

while the analysed signal is expressed as:

$$f(t) = \sum_{k=1}^{K} \tilde{d}_k(t) + R(t) \tag{8}$$

One of the limitations of the CEEMDAN is the selection of the amplitudes of the $\omega^{(i)}(t)$; however, experts in the field suggest the use of large-amplitude values for low-frequency-dominated signals and vice versa [19].

### 3.1.2. SAE-Based Feature Learning and HI Construction

Like most deep neural networks (DNNs), a typical AE architecture consist of an input layer, hidden layer, and output layer whereby the hidden and output layers form the *encoder* and *decoder*, respectively. Invariably, the encoder learns how to interpret the input and compresses it to an internal representation defined by the bottleneck layer while the decoder takes the output of the encoder (the bottleneck layer) and attempts to recreate the input.

Supposing the extracted modes $\tilde{d}_k(t)\epsilon R^d$ ($k = 1,...,K$ and $d$ is the dimension) are provided as inputs, the number of units in the input layer is $d$ and the encoding of the AE is obtained by a nonlinear transformation function using (9):

$$\mathbf{y} = f_e(\mathbf{W}\tilde{d}_k(t) + \mathbf{b}) = f_e(\widetilde{\mathbf{X}}) \tag{9}$$

where $y\epsilon R^h$ represents the hidden layer's output—*code*(feature representation). $h$ is the number of nodes in the hidden layer while $W\epsilon R^{hxd}$ is the input-to-hidden weights. $f_e$ is a nonlinear activation. The *ReLu* activation function is quite robust for regression/forecasting problems due to their superior advantage over other popular activation functions—*softmax*, *sigmoid*, and *tanh*, for avoiding vanishing gradient problems. This is defined in (10) as:

$$f_e(\widetilde{\mathbf{X}}) = \max[0, f_e(\widetilde{\mathbf{X}})] \tag{10}$$

Furthermore, the ReLU activation function ensures a comprehensive learning by the encoder to achieve a a reliable deep feature extraction for prognostics. As shown in Figure 1, stacking AEs to create SAEs further provides an avenue for high-level feature extraction from which a single comprehensive HI can be generated at the SAE's bottleneck.

Assuming there are $M$ hiden layers in the encoding part, the outputs at the $n$th encoding layer is computed using (11):

$$\mathbf{y}^{(n+1)} = f_e\left(\mathbf{W}^{(n+1)}\mathbf{y}^{(n)} + \mathbf{b}^{(n+1)}\right), \quad n = 0,\ldots,M-1 \tag{11}$$

where $\mathbf{y}^{(0)}$ is the input $\widetilde{\mathbf{X}}$ while $\mathbf{y}^{(M)}$ is the output of the last encoding layer—the high-level features.

For the decoding part, the output of the first decoding layer constitute the input of the second decoding layer and so on. These decoder outputs at the $n^{th}$ layer is computed using the $f_d$-activated output shown in (12).

$$\mathbf{z}^{(n+1)} = f_d\left(\mathbf{W}^{(L-n)T}\mathbf{z}^{(n)} + \mathbf{b}'^{(n+1)}\right), \quad n = 0,\ldots,M-1 \tag{12}$$

where the input $\mathbf{z}^{(0)}$ of the first decoding layer is the output $\mathbf{y}^{(M)}$ of the last encoding layer which in turn, constitutes the reconstructed input—$\widehat{\mathbf{X}}$.

To construct the health indicator (HI) for monitoring, the bottleneck's dimension is set to a unity dimension thereby producing a single vector—HI—from which different health states can be monitored via a single comprehensive indicator. Invariably, the SAE serves as both a deep feature learner and feature fusion tool for comprehensive HI construction.

### 3.1.3. Early Warning Threshold/TSP Determination

Early warning ensures that maintenance is scheduled early enough before a complete equipment failure occurs; however, determining the right point in time when such alarms—time to start prediction (TSP) should be set remains one of the few challenges in building a reliable prognostics scheme. Ideally, such determination is dependent on the analysts/engineer's expertise and knowledge in the domain; however, due to stochasticity, uncertainties, and complexities associated with unsupervised learning processes and the need for real-time applicability, such a biased human judgement remains questionable. On the bright side, statistical principles provide a more reliable solution whereby the standard deviation proffers an intuitive paradigm for TSP determination. As proposed in [26], when the HI deviates from 5 times the standard deviation—$5\sigma$ of the HI at healthy state, it indicates an incipient fault has occurred thereby triggering the RUL prediction process.

### 3.2. GRU-Based Prognostics

As a significant improvement on the traditional RNNs and LSTMs, GRUs use fewer parameters, consume lesser computational resource, and train faster than their counterparts—traditional RNNs and LSTMs. In addition, their ability for learning temporal long-term dependencies from sequential data make them robust for prognostics and RUL estimation [23]. Similar to LSTMs, GRUs function via a gating mechanism but without an output gate. As Figure 2 shows, instead of the output gate found in LSTMs, a typical GRU architecture uses an *update* gate for information flow control into the memory and a *reset* gate for information control out of the memory.

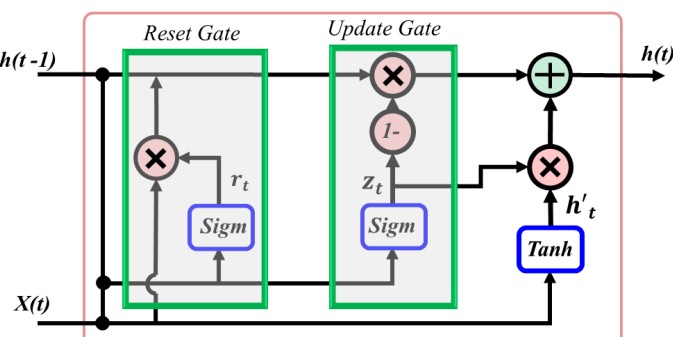

**Figure 2.** A GRU cell.

In simple terms, the reset gate learns the portion of the input data that needs to be forgotten while the update gate learns what portion of the input data that needs to be updated with newer data from the input. These enable for solving vanishing gradient problems and handling smaller datasets.

Equations (13) and (14) define the reset and update gates, respectively.

$$z_t = \sigma(W_z * [h_{t-1}, x_t]) \tag{13}$$

$$r_t = \sigma(W_r * [h_{t-1}, x_t]) \tag{14}$$

where $W_z$, and $W_r$ are their weight matrices, respectively, while $\sigma$ is a *Sigmoid* activation function.

The current state $h(t)$ is generated from $h'_t$ and an element-wise multiplication of the previous memory $h(t-1)$ and the update gate where $h'_t$ is the *Tanh*-activated output of the of prior output $h(t-1)$. They are defined in (15) and (16) as:

$$\tilde{h}_t = \tanh(W * [r_t * h_{t-1}, x_t]) \tag{15}$$

$$h_t = (1 - z_t) * h_{t-1} + z_t * \tilde{h}_t \tag{16}$$

In this study, the GRU-based prognostic module is triggered for RUL estimation when the fault threshold is reached.

## 4. Experimental Case Study

This section presents a practical case study whereby the proposed CEEMDAN-assisted RUL estimation technique is employed on a run-to-failure experiment on a VSC63A5 solenoid Pump produced by Korea Control Limited.

### 4.1. Testbed Description and Data Acquisition

In previous studies on failure diagnostics of electromagnetic pumps [4,27,28], filter clogging has shown to be one of the high-ranking failure modes. In this study, a natural running condition (powered by 220 V, 60 Hz) was simulated for a VSC63A5 solenoid Pump whereby the working fluid—5 Litres of diesel was contaminated with ten(10) grams of Iron(III) oxide ($Fe_2O_3$). Figure 3 shows the experimental setup showing with a picture of the physical test-bed and an illustration for the for sensor placements and data acquisition.

**(a)**

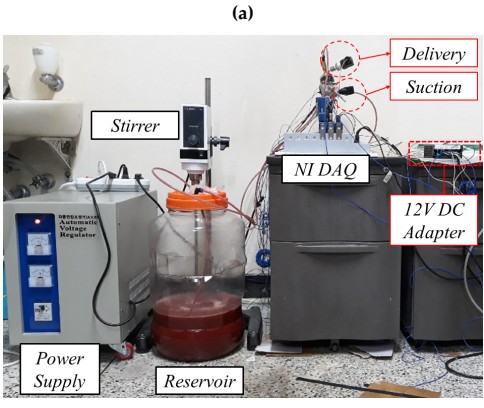

**(b)**

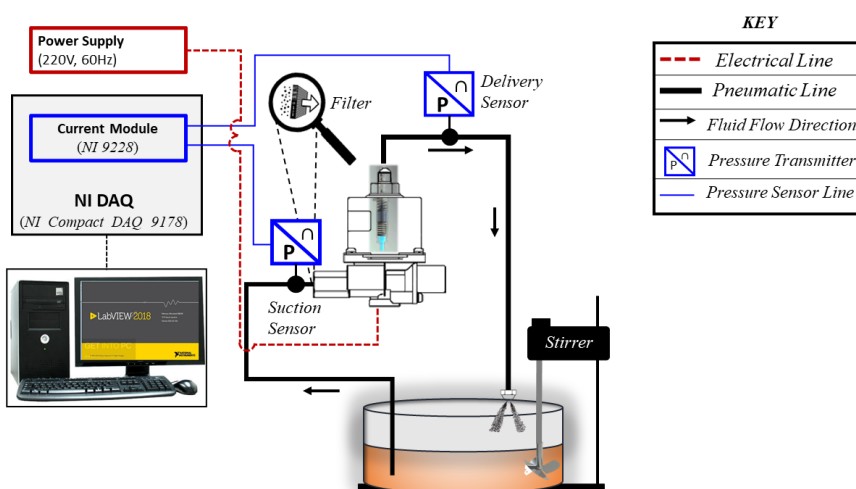

**Figure 3.** Experimental setup showing (**a**) A picture of the actual testbed (**b**) illustrations for sensor placements and data acquisition.

As the pump is powered, the oscillatory motion of the plunger induces a suction process from which fluid is transferred from the reservoir and discharged through the pump's delivery port (via a 1.0 GPH nozzle) back to the reservoir. A stirrer is installed for contous brownian movement of the contaminants while the pressure measurements

are the collected digitally from the suction and delivery ports at 20 KHz via two WIKA A10 transmitters (powered by a 20 V DC adapter) connected to an NI 9228 current module. The module was connected to a NI Compact DAQ 9178 data acquisition system which provided the digital signals to a PC through a LabView environment and stored in ".csv" file format.

### 4.2. Experimental Results and Observations

The experiment lasted for about 2448 h (102 days) with a resultant reduction in flowrate from the nozzle. Upon observation, it was revealed that the suction filter had been fully clogged. Figure 4 shows the different clog stages throughout the experiment.

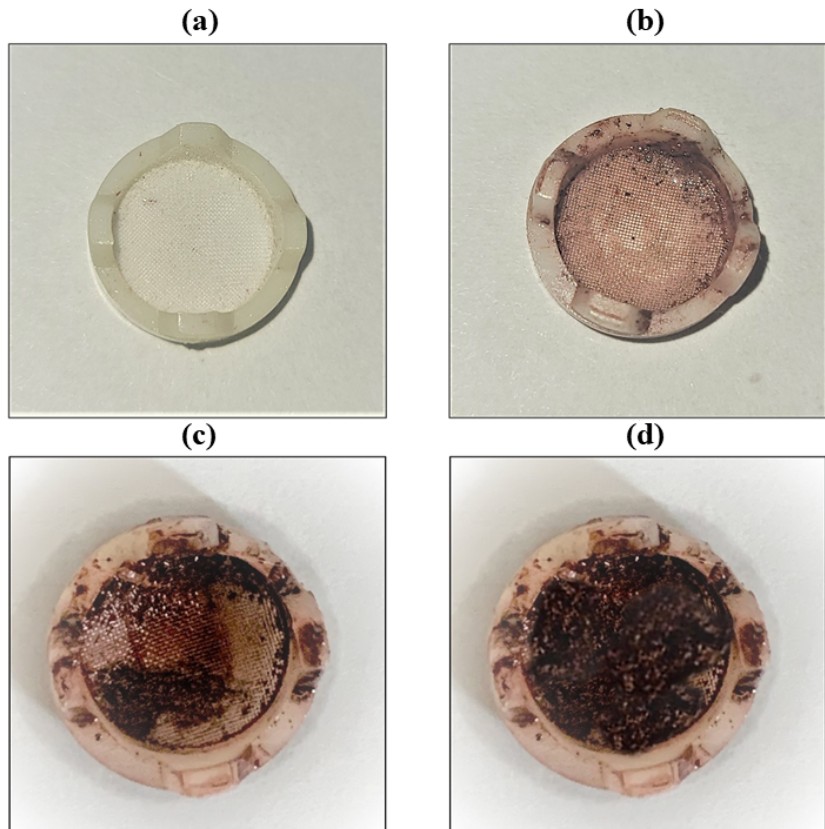

**Figure 4.** Pictures showing the suction filter at different stages (**a**) before the experiment, (**b**) normal running condition, (**c**) partial clogging leading to cavitation, and (**d**) clogged filter at end of experiment.

For the first 1100 h (first 46 days) of operation, the pump's *suction − delivery* process was observed to be steady/ideal even though some (negligible) sedimentation on the filter was observed (see Figure 4b). Beyond the 46th day, bubbles were observed in the transparent pipes with more fluctuations in the pressure signals signalling a drop in the pump's performance (see Figure 4c). This behaviour continued till the end of the experiment, when there was little/no fluid delivery from the pump due to full filter clogging as shown in Figure 4d.

### 4.3. CEEMDAN-Based De-Noising

Figure 5 shows the pressure signals acquired from the sensors for the whole run-to-failure time.

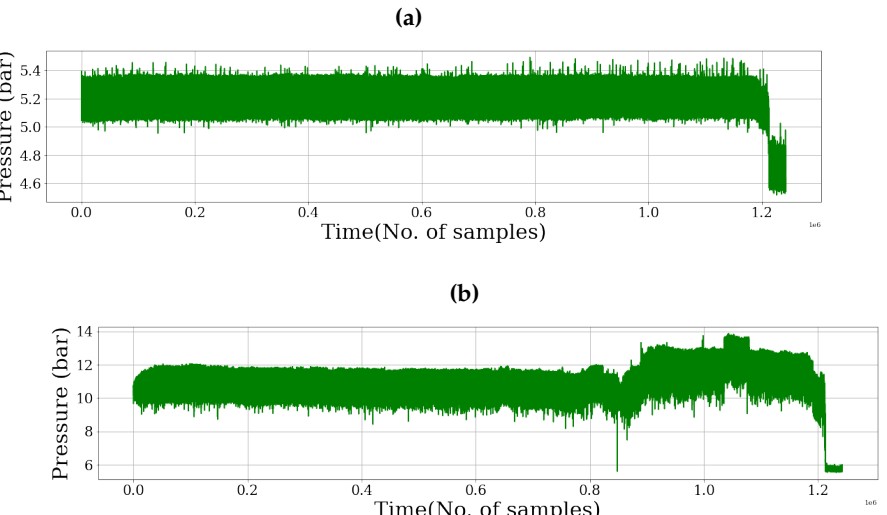

**Figure 5.** Pressure signals for the whole run-to-failure test (**a**) Suction, (**b**) Delivery.

As observed in Figure 5, the time-series signals contain high frequency components which are most probably from noise and uncertain sources. The CEEMDAN algorithm was employed, respectively, on the suction and delivery pressure measurements to produce the IMFs shown in Figure 6 whereby the IMFs in red are from the suction pressure signal whereas the IMFs in blue are from the delivery pressure signal.

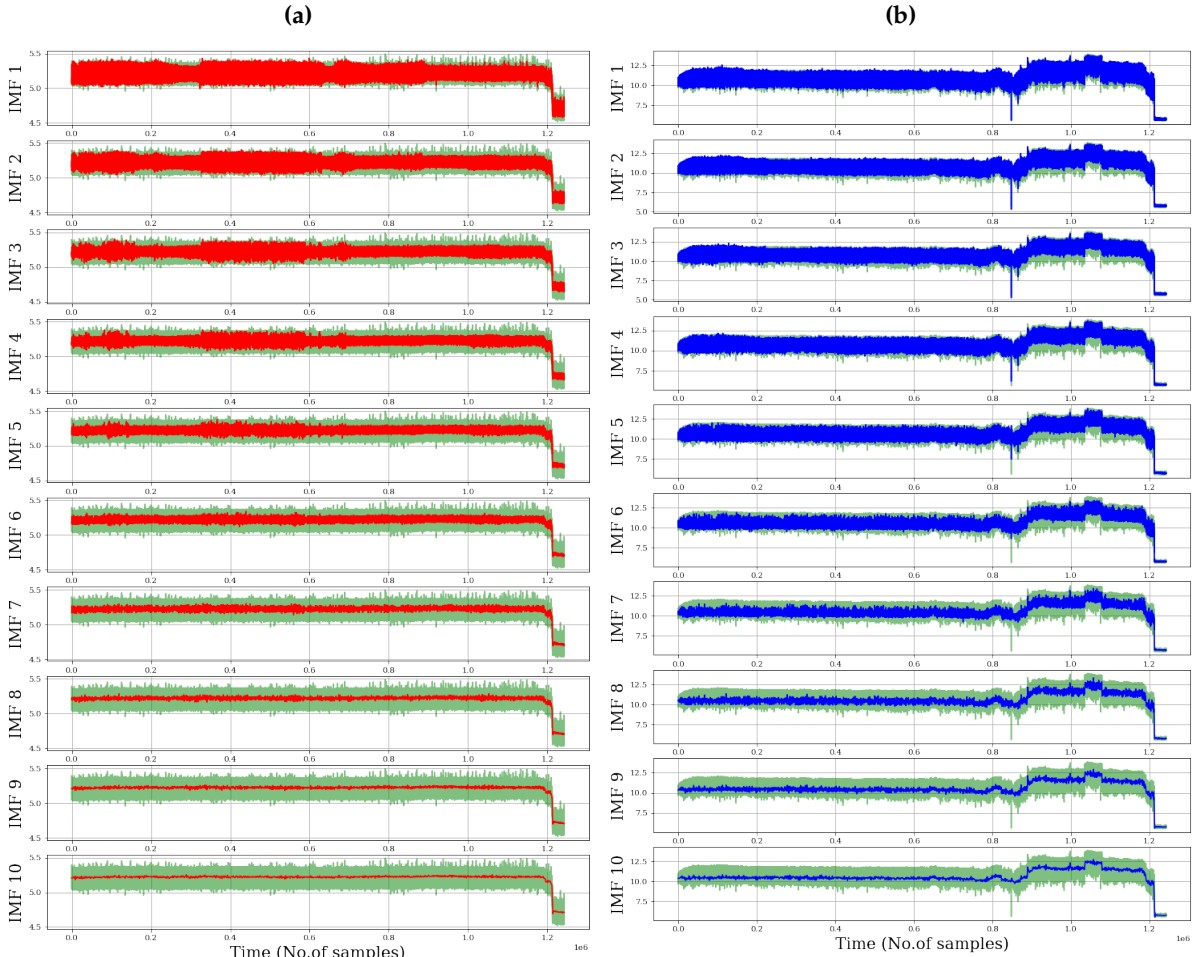

**Figure 6.** Extracted IMFs from the pressure signals from the whole run-to-failure test (**a**) Suction, (**b**) Delivery.

An ensemble value of 100 was set while the shifting iterations ranged between 15 to 20 to compute one IMF. The added white Gaussian noise was set to have a maximum amplitude of 0.25 of the original signals' standard deviation. Consequently, 20 IMFs were extracted from the suction and delivery pressure measurements, respectively (10 from each sensor measurements), and concatenated as inputs for feature learning. As observed in the differences between the IMFs and the raw signals (in green), the CEEMDAN algorithm efficiently output the relevant finite IMFs with practical significance for feature learning and RUL estimation.

### 4.4. HI Construction and TSP Determination

Using the 20 IMFs as inputs, the SAE-GRU prognostics model whose architecture is summarised in Table 1 was employed. First, the SAE accepts the inputs via the input layer of *Encoder 1* which then extracts the low-level features from the inputs. From these low-level features, *Encoder 2* further extracts the high-level features which are received via a *ReLU* activation function at the bottleneck for efficient learning. Being that the SAE's goal is to fuse the 20 IMFs to a single comprehensive HI, our chosen SAE architecture was motivated by implementing a two-step dimensionality reduction process via the encoders in a realistic fashion—$\{20 \rightarrow 10 \rightarrow 5 \rightarrow 1\}$. On the other hand, the GRU parameters were chosen based on multiple trials on different parameter architectures and experience in the domain.

As observed in Figure 1 and Table 1, the output dimension at the bottleneck is set to *1*. This implies that at this level in the SAE model, a single vector—comprehensive HI is returned, from which different health states can be identified along the time-series.

**Table 1.** Configuration of the proposed SAE-GRU architecture.

| Layer | Architecture | Description |
|---|---|---|
| Input | 20 | The dimension of the IMFs |
| Encoder 1 | 10, *B_norm* | Number of output nodes: 10, Batch Normalization, Activation: ReLU |
| Encoder 2 | 5, *B_norm* | Number of output nodes: 5, Batch Normalization, Activation: ReLU |
| Bottleneck | 1 | Number of nodes: 1 |
| Decoder 1 | 5, *B_norm* | Number of output nodes: 5, Batch Normalization, Activation: ReLU |
| Decoder 2 | 10, *B_norm* | Number of output nodes: 10, Batch Normalization, Activation: ReLU |
| Output | 20 | Number of nodes: 20, Activation: Linear |
| GRU 1 | 50 | Number units: 50, Activation: ReLu |
| Gaussian Dropout | 0.2 | 0.2 dropout |
| GRU 2 | 20 | Number units: 20, Activation: ReLu |
| Gaussian Dropout | 0.2 | 0.2 dropout threshold |
| Dense | 1 | Number of output nodes: 1, Activation: Linear |

Because the feature learning process of the SAE is unsupervised, its efficiency to produce a reliable HI depends on its efficiency to accurately produce the original inputs (IMFs) at *Decoder 2*'s output. This can be monitored/assessed via the error convergence of the training with the validation set from which a *zero-convergence* implies a reliable learning.

The results of the SAE learning process is shown in Figure 7 over 100 iterations which produced the HI shown in Figure 8.

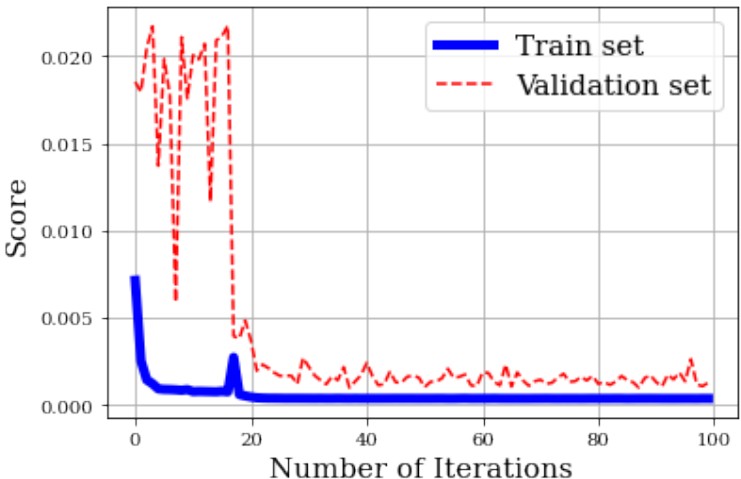

**Figure 7.** Training process of SAE over 100 iterations.

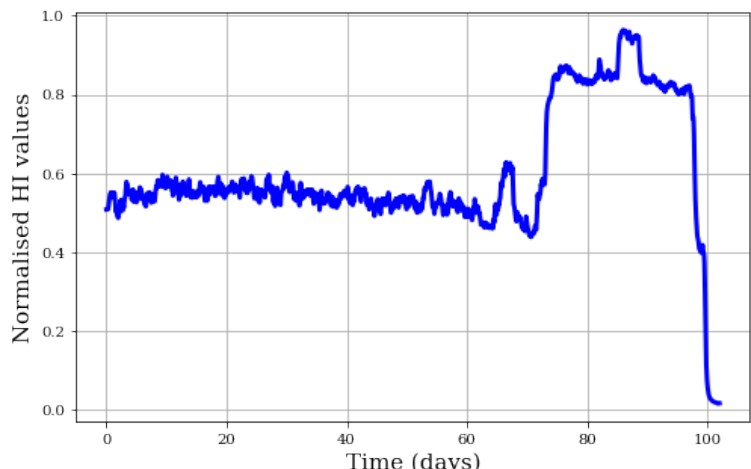

**Figure 8.** HI produced by the SAE for the whole run-to-failure experiment.

As shown, the error convergence of the SAE feature learner is quite impressive. This in return validates the reliability of the HI extracted from the SAE's bottleneck shown in Figure 8.

To determine the TSP using the $5\sigma$ technique, first the standard deviation ($\sigma$) of the signals healthy state was computed for the first 5000 samples of the time-series—HI in Figure 9. It is assumed that the early-life sensor measurements (i.e., for the first few days) reflect the healthiest state of the pump.

Consequently, an averaged $\sigma$ value of 0.01 was obtained from the samples at healthy state. Figure 9 also shows the windowed output of the $\sigma$ technique across the HI from left (healthy) to right (fault/failure) (in green lines). As shown in the red dotted horizontal lines, the TSP falls at about day 68 whereby the $\sigma$ value of 0.05 ($5 \times 0.01$) was first returned. This was actually within the period where bubbles were observed in the transparent fluid lines (cavitation) signalling an incipient fault stage due to partial filter clogging. A closer look at Figure 9 would also reveal the early warning efficiency of the $5\sigma$ technique for improved real-time monitoring. After day 68, it is shown that the standard deviation of the HI returned to an acceptable range til day 73 where an even higher increase in the standard deviation is observed. By human observation of the HI, one may assume the pump is still in its normal running state at day 68 without realising that in just a few more days—from

day 73 (marked by a *red circle*), the pump's running condition would become even worse and it may be reaching much closer to its end of life. This validates the need for the $5\sigma$ as an early warning metric for reliable condition monitoring.

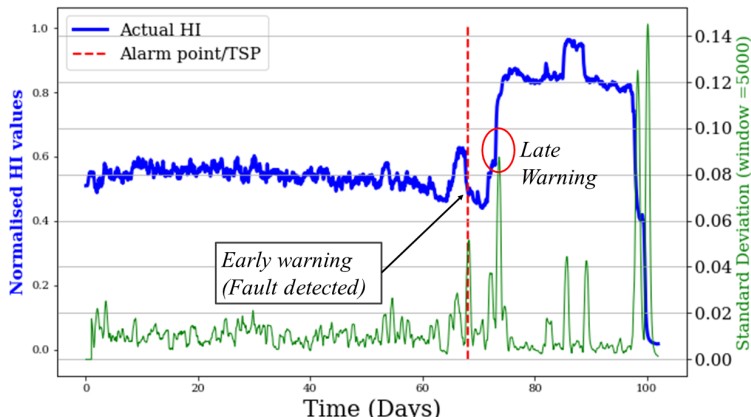

**Figure 9.** HI illustrating the TSP from the $5\sigma$ technique.

*4.5. GRU-Based RUL Estimation*

The developed model returns RUL estimates as outputs when provided the processed data. Since the anomalous behaviour was detected on day 68, the actual RUL (34 days) decreases from this point to end-of-life (EOL) (day 102) and forms the ground truth data for assessing the proposed model's performance. As shown in Figure 9, it is assumed that the failure threshold is at minimum HI value (HI = 0).

4.5.1. GRU Initialization and Training, and Validation

When a fault alarm is triggered at TSP (when the $5\sigma$ threshold is reached), the GRU predictor accepts the HI from healthy state to the TSP as input for training for making posterior RUL estimates based on the learned input data. The RUL till this point is assumed to be at 100% (no fault witnessed); beyond which the equipment fails gradually to 0% RUL (end-of-life). The training process is supervised with the HI from day 0–day 68 mapped to the constant 100% as output for training. This trained model is then deployed for making RUL predictions based on the new (fault) data. Consequently, considering the stochastic nature of the GRU and the need for optimal minimal false alarm rate, a 3-layer GRU network with *ReLU* activation function between layers was designed for improved learning whereas at the output node, a *linear* activation function was selected (with *adam*) optimization for model stability.

The hardware used has the specification summarised in Table 2 while the computation process was done in *Keras* with *Tensorflow* back-end. The batch size for each iteration was set to 1024 and was run over 500 iterations with mean square error (MSE) as the loss function. Figure 10 shows the result from the training process over the 500 iterations with minimal training/validation error convergence (towards zero) using the architecture summarised in Table 1.

**Table 2.** Specification of computational hardware used.

| Manufacturer | Processor | Speed | RAM size |
|---|---|---|---|
| Advanced Micro Devices (AMD) | Ryzen 7, 2700 Eight-core | 3.20 GHz | 16 GB |

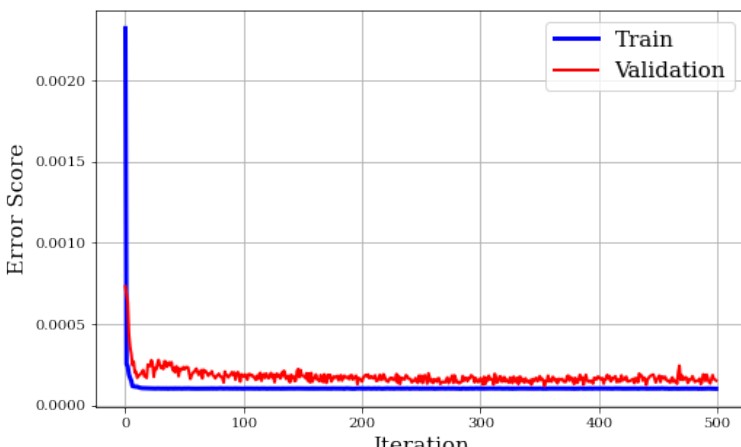

**Figure 10.** HI illustrating the TSP from the 5σ technique.

The validation convergence in Figure 10 over the iterations provides a reliable insight on the successful learning by the model and more invariably, the predictive capacity of the trained model for posterior (RUL) estimations in an unsupervised manner. At TSP, the trained GRU model predicts the RUL till EOL from which the estimated RUL values are compared with the ground truth data for performance evaluation; however, to ascertain the predictive capabilities of the model for RUL estimation, we first tested the model's self-learning efficiencies by predicting the HI input values from TSP to EOL. Next, the self-learning efficiency of the model further provides reliable insights on its effectiveness for making RUL estimates given these test HI values (from TSP to EOL). Figure 11 shows the one-step ahead prediction result by the GRU estimator for from TSP to EOL whereby the actual HI degradation trend is represented in blue lines while the predicted HI degradation trend is represented in the red lines.

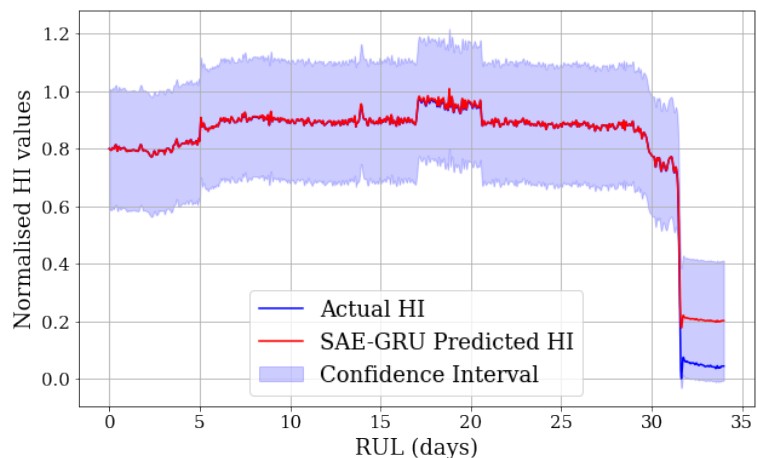

**Figure 11.** One-step ahead prediction by GRU estimator.

As shown, the model returned an identical degradation trend confidently (within 95% confidence interval) bounded by the light blue area. As expected, the estimator was less confident towards the EOL since there was a sharp degradation trend towards the EOL. The authors are confident that this sharp drop towards the EOL of the pump is associated with the power surge experienced during the experiment and may have impacted on the actual degradation trend of the pump; nevertheless, the confidence of the GRU predictor remains reliable for real-time applications where little/no abrupt changes in operating conditions are controlled/prevented.

4.5.2. RUL Estimation Results and Discussions

With the trained model, the RUL estimation from TSP to EOL was predicted and visualized using the $\alpha - \lambda$ metric—a reliable offline prognostic evaluation tool that outputs either 1 or 0 for predictions within a cone of accuracy–$\alpha$ bounds, at a specific time index–$\lambda$ [29]. The easy representability, comprehensibility, and visual-friendly features of the $\alpha - \lambda$ metric it not only popular, but also very reliable for visualizing, evaluating, and reporting the RUL estimation performance of a prognostic model. Equation (17) provides the mathematical definition of the $\alpha - \lambda$ metric.

$$\alpha - \lambda \text{ Accuracy } = \begin{cases} 1 & \text{if} & \pi[r(i_\lambda)]_{-\alpha}^{+\alpha} \geq \beta \\ 0 & \text{otherwise} \end{cases} \tag{17}$$

where $t_\lambda$ is a fraction of time between TSP and the actual EOL, $\lambda$ is the time window modifier such that $t_\lambda = t_P + \lambda(t_{EoL} - t_P)$, $\beta$ is the minimum acceptable probability for $\beta$ criterion, $r(i_\lambda)$ is the predicted RUL at time index $i_\lambda$, and $\pi[r(i_\lambda)]_{\alpha^-}^{\alpha^+}$ is the probability mass of the prediction PDF within the $\alpha$-bounds that are given by $\alpha^+ = r_*(i_\lambda) + \alpha \cdot r(i_\lambda)$ and $\alpha^- = r_*(i_\lambda) - \alpha \cdot r(i_\lambda)$

The RUL estimation result by the proposed model (visualized using the $\alpha - \lambda$ metric with $\alpha$ = 0.3) is presented in Figure 12.

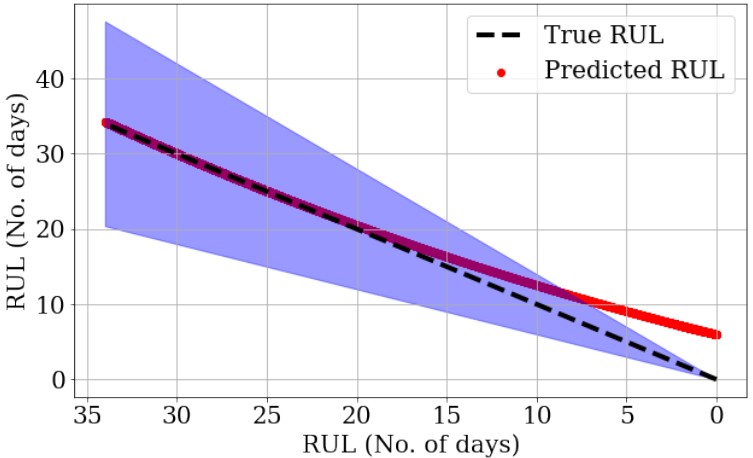

**Figure 12.** RUL prediction results by GRU at TSP (68th day).

As shown, the model's predictive accuracy for the actual RUL (34 days) falls within acceptable $\alpha$ bounds until about the last 8 days where the prediction falls outside the $\alpha$ bounds. As earlier seen in Figure 11 where the prediction for the last few days are observed to be less confident due to the sharp degradation trend, the RUL prediction results in Figure 12 better reflects this error/anomaly towards the EOL.

*4.6. Performance Evaluation*

Apart from sensor line malfunction, some sources of error that could be associated with a PHM system include data noise, observer faults, etc. These factors play vital roles on choosing the appropriate prognostic metrics to adopt for evaluations which may include logistics, saftey, reliability, mission criticality, and economic viability; however, to better assess the accuracy-based prognostic performance of a model, it is often wiser to explore metrics like the root mean square error (RMSE), mean absolute error(MAE), relative error (RE), etc. especially for uniformly comparing certain aptitudes or measures across several algorithms. Comprehensively, these metrics when incorporated into decision making processes better provide reliable standpoints for developing trust in a a prognostic model.

The complexity and predictive performance of the developed model was evaluated with the earlier listed prognostics evaluation metrics in comparison with other ML-based estimators—multi-objective genetic algorithm-optimized long short term memory (MOGA–

LSTM) [4], deep belief network (DBN) [22], and a 3-layer deep neural network (DNN) which had been earlier deployed/developed for the same purpose in our past study. It is worth noting that these algorithms shared a similar architecture like the proposed method for fair comparison since it would be futile to explore/optimize different variants/architectures for the respective models. Each of these models, like the developed model, receives the HI from the SAE's bottleneck to estimate RULs followed by a comparison in their performance. Due to the random weight initialization process of the algorithms, their respective performances were evaluated by retraining each of the models five times and computing their respective averaged errors. Their respective RUL predictive performances at TSP are compared in Figure 13.

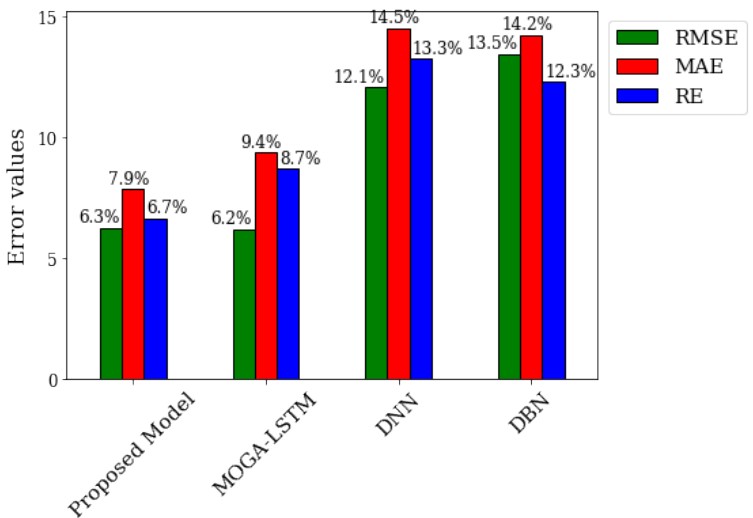

**Figure 13.** Performance comparison for RUL estimators at TSP.

As shown, the difference in the RMSEs, MAEs, and REs between the proposed model and the other estimators are quite significant. The MOGA-LSTM estimator reveals a strong competitive ability (and may perform even much better in some applications). This is associated with the MOGA optimizer in its architecture which optimizes the LSTM predictor. With continuous-valued stochastic units, the MOGA-LSTM retains a strong efficiency in handling the input variables (HI) with complex non-linear characteristics; however, due to the computational costs associated with its use, the proposed model presents itself as a cost-efficient option. On the other hand, the DBN and DNN estimators; although reliable, performed the least across the evaluation criteria presented. It is projected that the DBN and DNN may be unreliable under different unseen operating conditions and/or poor parameterization which can be traced to the vanishing gradient problems that they are popularly prone to. In essence, the proposed model comes with a much more reliable architecture, cost-efficiency, ease-of-use, better generalization capabilities (due to automated feature learning), fewer parameterization, durability, and real-time applicability.

## 5. Concluding Remarks

Data-driven predictive maintenance relies significantly on the availability and prognosibility of sensor data from which the linear and nonlinear characteristics of the target system are identified for a comprehensive dynamic modelling. Signal de-noising is almost a necessary pre-processing step to ensure that noise and/or insignificant spectral components of the signals are excluded from in the feature extraction and dynamic modelling process. The EMD variant—CEEMDAN offers a solution by generating fewer IMFs on the premise of separating different signal components which provide reliable inputs for health indicator construction by SAEs. GRUs, on the other hand, whose prowess for learning long-term dependencies ensures a highly reliable RUL estimation. The CEEMDAN-assisted prognostics model proposed in this study relies on the SAE for feature learning and HI

construction and the GRU for RUL estimation. This was tested on a run-to-failure operational data from a VSC63A5 Solenoid pump produced by Korea Control Limited under an ideal operating condition with an $Fe_2O_3$-contaminated diesel as the operational fluid and its performance compared with other effective RUL estimators. Results are supported by extensive empirical validations with the proposed method revealing better cost-efficiency, minimal false alarm rate, and ease-of-use.

The developed model can be deployed on any system that utilizes pressure (or non-stationary) signal analysis; however, its efficiency may be limited for much more complex systems which may require a more exhaustive search for the right SAE configuration for reliable HI construction. As the overall neural network architecture deepens, the non-linearity/complexity between inputs and target variables increase and this may have inhibiting effects on predictive accuracies. Continued research shall be aimed at obtaining more experimental data to cover other failure modes for a more comprehensive prognostic scheme with the hopes of validating the efficacy of the proposed model.

**Author Contributions:** conceptualization, U.E.A.; methodology, U.E.A.; software, U.E.A.; formal analysis, U.E.A.; investigation, U.E.A.; resources, U.E.A. and J.-W.H.; data curation, U.E.A.; writing-original draft—U.E.A., writing—review and editing, U.E.A. and J.-W.H.; visualization, U.E.A.; supervision, J.-W.H.; project administration, J.-W.H.; funding acquisition, J.-W.H. All authors have read and agreed to the published version of the manuscript.

**Funding:** This research was supported by the MSIT(Ministry of Science and ICT), Korea, under the Grand Information Technology Research Center support program(IITP-2020-2020-0-01612) supervised by the IITP(Institute for Information & communications Technology Planning & Evaluation).

**Data Availability Statement:** The data presented in this study are available on request from the corresponding author. The data are not publicly available due to laboratory regulations.

**Conflicts of Interest:** The authors declare no conflict of interest.

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
