# Peer review of "A CEEMDAN-Assisted Deep Learning Model for the RUL Estimation of Solenoid Pumps"

_electronics, doi:10.3390/electronics10172054_

Round 1

Reviewer 1 Report

The manuscript is well detailed and presented. However, some minor issues arise:

  1. Lines 18-19. ML techniques are highly data dependent (with large quantities of data necessary to achieve good results) and their determination requires high performance equipment. Thus, the affirmation “however, ML techniques retain their superiority on small data and low-end hardware” is confusing. Please clarify this aspect and support it with references.
  2. Lines 22-23. Please verify English language. Similar for lines 51, 79, 87-88, 124-125, 130, etc…
  3. Please specify all notation before using them. Line 33 (SAE), line 97 (AE)
  4. Line 176. To eliminate confusions, please also mention the producer of that specific pump
  5. Table 1. Based on what criteria the parameters were selected? I.e. Why Relu, with 10 outputs nodes for encoder 1, etc…?
  6. Placement of figures. Figs 1-6 are in-text and Figs 7-9 are at the end of the manuscript. Please be consistent regarding Figure placement. Also. Fig 3,7 are incomplete. Please format them to fit the page.

Reviewer 2 Report

This paper discussed an interesting topic, namely the RUL estimation of solenoid pumps. The CEEMDAN, SAE and GRU are combined and applied. The experiment seems to verify the effectiveness of this method. However, the procedure and method description are not clear. Before acceptance, the authors should make a major revision by addressing the following comments.

  1. Please check the writing carefully. The quality of the figures is poor. For instance, parts of the figure 3 and figure 7 are missed. Please insert the figures into the words. It is difficult for readers to check the manuscript.

  1. The author said “In this study, using a window size of 5000 samples across the time-series– HI, an averaged standard 240 deviation value of 0.01 was returned from the samples at healthy state.” For each sample in the blue line of figure 10, is it in the center, left or right of the window? The reviewer thinks the deviation value should be calculated among the previous 5000 samples with respect to current point.

  1. The reviewer is confused about the GRU-based RUL prediction. The input of GRU is a single value, namely the each data point of HI value in figure 10? Then, what are the outputs of GRU? The GRU is suitable to analyze the time-series, rather than a single data point. If so, the simple MLP can handle this problem.

  1. The performance evaluation of RUL process should be presented as figure 1. The figure 12 is strange here. What are the meanings of these lines? The authors said HI values. Then, what are the relationships between figure 12 and figure 9? Besides, from the words “Since the anomalous behaviour was detected on day 73, the actual 255 RUL decreases from this point to EOL (day 102) and forms the ground truth data for assessing the 256 proposed model’s performance.”, the output of GRU should be the remaining days to failure. However, from the words “Having created the HI, the GRU predictor is designed to accept this HI as input for training; however, the training process in done with only the healthy data (from day 0 – day 68).”, then what are the output labels of the training data? The healthy data (from day 0 – day 68) may have no corresponding remaining days to failure. The authors should re-write this part.

  1. The detailed network parameters of the comparative methods, including DNN, DBN, MOGA-LSTM, are also unknown? How to ensure the best results achieved for a fair comparison in these methods?

  1. The generalization ability of trained model is significant in practical engineering. The proposed methods can be robust to the change of working conditions or unseen condition during the training of model. The current experiments may be conducted in a single condition, indicating the single speed and procedure. The authors may give more descriptions in this point by referring to: "A hybrid generalization network for intelligent fault diagnosis of rotating machinery under unseen working conditions," in IEEE Transactions on Instrumentation and Measurement, 2021, 70: 3520011; Mechanical Systems and Signal Processing, 2019, 117: 170-187.

Reviewer 3 Report

The introduction presents a large number of references that show a good study of the problem analyses.

 The methods are connected with analytical formulas for a good understanding of analysis realize it.

The conclusion and results are connected and present the dates measurements but it will need to be improved with numbers and percents of dates.

,,As shown, the difference in the RMSEs, MAEs, and REs between the proposed model and the other estimators are quite significant. " - will be good to put the percentage of which means significant like example.

Round 2

Reviewer 2 Report

The authors have addressed all my comments. The reviewer would like to recommend the "Acceptance".